# ADHD Symptoms Increase Perception of Classroom Entropy and Impact Teacher Stress Levels

**DOI:** 10.3390/children10061082

**Published:** 2023-06-20

**Authors:** Rosa Angela Fabio, Carmela Mento, Antonio Gangemi, Giulia Picciotto

**Affiliations:** 1Department of Economics, University of Messina, 98122 Messina, Italy; 2Department of Biomedical and Dental Sciences and Morphological and Functional Images, University of Messina, 98122 Messina, Italy; carmela.mento@unime.it; 3I.R.C.S.S. Bonino Puleio, 98100 Messina, Italy; biosneurolab@outlook.com; 4Madonna della Consolazione Polyclinic Nursing Home, 89124 Reggio, Italy; 5Department of Clinical and Experimental Medicine, University of Messina, 98122 Messina, Italy; giulia.picciotto@hotmail.com

**Keywords:** children with ADHD, psychological entropy, teachers’ stress, teacher–student relationship

## Abstract

Attention deficit hyperactivity disorder (ADHD) is a common neuropsychiatric disorder among school-age children, characterized by persistent behavioral patterns of inattention and/or hyperactivity/impulsivity. These behaviors can create stress for teachers and negatively affect teacher–student interactions. This study hypothesized that a high frequency of ADHD students in the classroom can increase internal and external entropy, ultimately resulting in a negative stress impact on teachers. The physical concept of entropy, which measures the degree of disorder in a system, was used to better understand this relationship. The study evaluated 177 primary school teachers in their response to interacting with students with ADHD, using the Measurement of Psychological Stress (MPS) to evaluate subjective stress levels and the QUEIs and QUEIp questionnaires to measure structural and personal entropy. Path analysis was applied to identify the factors associated with the total score of MPS. The hypothesis was confirmed, as the frequency of ADHD students had a negative impact on teachers’ entropy levels and personal entropy was found to significantly increase stress levels. The study highlights the negative impact of ADHD symptoms on stress levels and personal entropy of teachers when interacting with students with ADHD. These findings suggest the need for interventions aimed at balancing the frequency of students with ADHD and promoting positive training on stress reduction for teacher–student interactions.

## 1. Introduction

The significance of the teacher–student relationship in the social and academic development of students has been extensively documented in various studies [1,2,3,4]. It is widely recognized that the teacher–student relationship is a fundamental prerequisite for effective learning [4]. Furthermore, teachers’ ability to responsively address students’ needs and requirements is closely linked to student performance [5]. However, problematic student behavior can significantly contribute to teacher stress [6]. Numerous studies have demonstrated that teachers experience higher levels of job stress compared to professionals in other fields [7,8]. One significant stressor for teachers is student misbehavior [9,10], which becomes particularly distressing when it escalates to aggressive behavior [11]. Aggressive behaviors are indeed within the symptomatic framework of attention deficit hyperactivity disorder (ADHD). However, only a few studies have specifically focused on the teacher–student relationship in the context of students with ADHD [12], and how this disorder significantly impacts the quality of that relationship [13,14], thereby compromising the well-being of both teachers [15] and students with ADHD [12].

ADHD is one of the most prevalent neuropsychiatric disorders, especially among school-age children [16]. According to the DSM-5, ADHD is a neurodevelopmental disorder characterized by persistent and pervasive patterns of inattention and/or hyperactivity/impulsivity [17]. Children with ADHD often struggle with maintaining attention, controlling impulses, regulating activity levels, and engaging in proper behavior [18]. The disorder’s etiology involves various contributing factors, including neurobiological origins that interfere with a child’s normal psychological development, as well as numerous educational and environmental triggers [19,20,21,22]. ADHD symptoms tend to have negative effects on interpersonal relationships [23]. Students with ADHD create a disruptive environment within the classroom, interrupting teaching and learning processes and distracting both teachers and classmates. They may frequently blurt out answers without waiting to be called upon, exhibit motor restlessness [24], experience slower cognitive processing [25], have decreased attention spans [26], and engage in conduct problems, hyperactivity [27], or verbal and physical aggression towards classmates [28,29,30,31,32]. The response of adults to hyperactive behavior in children with ADHD is highly subjective, and the level of stress experienced can be influenced by individual perceptions of the event [33]. 

This study incorporates the concept of “entropy”, derived from thermodynamics and information theory, to gain insights into disruptive situations that may occur in the classroom. Entropy, in this context, refers to the measure of uncertainty and disorder within a system [34]. Self-organizing systems engage in continuous interaction with their environment and must adapt to changing circumstances to maintain an acceptable level of internal entropy [34]. Additionally, for an organism to survive, it must effectively dissipate its entropy into the environment. The principles of entropy and self-organization can be applied to various complex information systems, including the investigation of psychological phenomena [35,36,37,38]. Psychological entropy is inversely correlated with an individual’s capacity to effectively engage in comprehensive tasks and achieve rewards through purposeful perception and action [34]. It specifically pertains to a person’s tendency to become disorganized, chaotic, and confused in their psychological life. This concept is based on the understanding that the human mind naturally gravitates toward order and organization but can be influenced by external and internal factors that disrupt its equilibrium.

Psychological entropy can manifest in various ways, such as memory loss, emotional confusion, lack of motivation, or difficulty in making decisions. Psychological entropy can be triggered by various factors, including stress, depression, or anxiety [34].

In the school context, the concept of “structural entropy” and “personal teacher’s entropy” have been measured. Structural entropy refers to a set of physical variables that make up the school environment and somehow influence the behavior of students and teachers, such as classroom disorder, irregular schedules, unclear rules, and classroom noise. On the other hand, personal teacher’s entropy is linked to subjective aspects of the teacher, such as attitudes, behaviors, educational styles, and classroom management approaches [39].

In this study, the hypothesis proposed is that a high frequency of ADHD symptoms may be associated with perceived teacher entropy, both structural and personal, and that this, in turn, is correlated with teacher stress [18]. It is expected that an increase in entropy will lead to a higher perception of stress by the teacher. The null hypothesis for this study is that there is no association between a high frequency of ADHD symptoms and perceived teacher entropy, both structural and personal. Additionally, it states that there is no correlation between teacher entropy and teacher stress. In other words, any observed relationship between ADHD symptoms, teacher entropy, and teacher stress is due to chance or other factors, rather than a true causal or correlational relationship. 

## 2. Methods

### 2.1. Participants

Out of 372 questionnaires distributed to teachers, only 47% responded to the questions asked. Therefore, the final sample included 178 teachers (18 men, 156 women, and 4 people who did not complete this answer) aged between 30 and 60 years (M = 43.76, SD = 7.95). On average, the participants had been working as primary school teachers for 19.78 years (SD = 11.03). Specifically, 9.04% had been working for less than 11 years, 14.69% from 11 to 20 years, 50.28% from 21 to 30 years, 11.86% for more than 30 years, and 14.13% did not indicate how many years they had been working in the school. This study included a total of 2339 children, who were enrolled from 177 classes in primary public schools located in Milan and Messina and their respective provinces. The age range of the children varied from 6 to 10 years old. The sample encompassed all the children within each classroom, comprising both clinically normal children and children with clinical conditions.

All clinically significant children had received an official diagnosis from healthcare facilities in their respective regions. Among the sample, 4.98% exhibited symptoms of attention deficit hyperactivity disorder (ADHD), 1.2% displayed aggressive behaviors, 3.2% had language disorders, 3.6% experienced difficulties in mathematics, and 2% presented intellectual disabilities. In Italy, students with disabilities typically enroll in mainstream schools and attend regular sections and classes across all educational levels. The teachers were asked to complete a questionnaire, providing information on the number of students with ADHD and other disabilities present in their classrooms. Table 1 shows the descriptive statistics.

### 2.2. Materials 

In the present study, teachers filled the Classroom Behavior Survey to describe the prevalence of specific students’ issues. Moreover, the Italian version of Psychological Stress Measure (PSM) [40,41] was used to evaluate stress levels of the teachers. The Structural Education-School Entropy Questionnaire (QUEIs) [39] was administered to assess the level of structural entropy, while the Personal Teacher Entropy Questionnaire (QUEIp) [39] was used to assess personal entropy levels.

#### 2.2.1. Classroom Behavior Survey

The teachers were asked to complete a comprehensive information questionnaire known as the ‘Classroom Behavior Survey’ (CBS). In this survey, they were required to assess the prevalence of specific problem categories among students, namely aggressive behavior, language-related difficulties, mathematical difficulties, and intellectual disabilities (Table 2). Teachers were instructed to report only students who had received official diagnoses from specialized physicians, excluding any subjective opinions or beliefs regarding the children’s difficulties. The focus was on reporting the actual number of students who displayed these specific behaviors in their classroom.

#### 2.2.2. Measurement of Psychological Stress

Psychological stress was assessed using the Italian version of the Psychological Stress Measure (PSM) [40,41]. The PSM defines psychological stress as a multifaceted response of individuals to environmental demands. Each person exhibits unique responses to stressors. Unlike many stress assessments developed for psychiatric or clinical populations, the PSM aims to measure perceived stress in the general population. It has undergone standardization on a sample of healthy individuals in Italy. Additionally, the PSM does not employ an inferential approach to assess stress, but rather focuses on an individual’s subjective perception of feeling “under pressure” [42]. The scale comprises 49 items that capture an individual’s perception of their cognitive-affective, physiological, and behavioral state. Scores range from 49 to 196, with higher scores indicating greater perceived stress. The PSM demonstrates good psychometric properties, including a high internal consistency with a Cronbach’s alpha coefficient of 0.95 and a test–retest reliability of 0.80. The total score derived from the PSM serves as a comprehensive indicator of psychological stress.

#### 2.2.3. QUEIs: Structural Education-School Entropy Questionnaire

The Structural Education-School Entropy Questionnaire (QUEIs) is a measurement tool designed to assess the structural elements contributing to the entropy of the school environment [39,43,44,45,46]. The questionnaire consists of 17 items that investigate various aspects of the school’s structure, including the level of clutter in the classroom, the presence of irregular activities and schedules, and the absence of clear educational rules and routines. These items aim to capture the overall organization and orderliness of the school setting. The QUEIs questionnaire also collects information about the total number of pupils in the class and identifies those students who are perceived as lively and hyperactive. To detect the structural entropy in the school environment, Fabio et al. [39] created this questionnaire, which investigates the structural aspects of the school environment, complemented by a questionnaire regarding the personal, individual, and subjective aspects of teachers. In this tool, the teacher is asked a series of questions and is required to indicate on an ordinal scale the frequency (on five levels) with which certain characteristics related to the presence of order in the environment, stability in the organization of school life, and adherence to rules are observed. The overall entropy score is calculated by summing all the scores related to the various items, based on the responses provided by the teachers. 

The internal consistency of the QUEIs questionnaire was assessed using Cronbach’s alpha coefficient, which is a measure of scale reliability. The questionnaire exhibited strong internal consistency, yielding a Cronbach’s alpha coefficient of 0.89. This indicates that the items in the questionnaire are highly correlated and collectively measure the intended construct effectively. Pinelli and De Nitto [47] confirmed the psychometric properties, the internal consistency with Cronbach’s alpha coefficient was 0.87; the test–retest reliability was 0.82. Overall, the QUEIs questionnaire serves as a valuable tool for assessing the structural elements contributing to the entropy of the school environment, providing insights into the organization and orderliness of the classroom setting, as well as the numerosity of the classroom and the presence of hyperactive students [39,47].

#### 2.2.4. QUEIp: Personal Teacher Entropy Questionnaire

The Personal Teacher Entropy Questionnaire (QUEIp) [39] is a comprehensive measurement tool designed to assess various personal aspects of teachers. Comprising 27 items, the questionnaire investigates individual and subjective characteristics such as attitudes, behaviors, educational styles, self-discipline, and dispersiveness. It aims to provide a comprehensive understanding of the teacher’s personal attributes that may influence their instructional practices and classroom dynamics. The QUEIp questionnaire prompts teachers to self-assess their attention span, degree of hyperactivity, impulsiveness, self-discipline, and level of dispersiveness. These measures capture individual differences in these characteristics among teachers and shed light on how these personal attributes may impact teaching effectiveness and interactions with students. In order to evaluate the reliability of the QUEIp questionnaire, internal consistency was assessed using Cronbach’s alpha coefficient. The overall entropy score was calculated by summing all the scores related to the various items, based on the responses provided by the teachers. The questionnaire showed high internal consistency, with a Cronbach’s alpha coefficient of 0.89. This indicates that the items in the questionnaire are highly correlated and collectively measure the intended construct effectively. Pinelli and De Nitto [47] confirmed the psychometric properties, the internal consistency with the Cronbach’s alpha coefficient was 0.86; the test-retest reliability was 0.85. The QUEIp questionnaire is a valuable tool for assessing personal aspects of teachers, including attitudes, behaviors, educational styles, self-discipline, and dispersiveness. It provides insights into individual characteristics that may influence instructional practices and classroom dynamics. The high internal consistency further supports the questionnaire’s reliability as a measurement tool [39,47,48,49].

### 2.3. Procedure

The experiment was conducted following the principles outlined in the Declaration of Helsinki. After obtaining written informed consent from the teachers, they were requested to complete the Classroom Behavior Survey to describe specific aspects related to the students’ issues, the Italian version of Psychological Stress Measure (PSM), the Structural Education-School Entropy Questionnaire (QUEIs) and the Personal Teacher Entropy Questionnaire (QUEIp). The order of presentation of the questionnaires was randomized.

### 2.4. Statistical Analysis 

Descriptive statistics, such as means (SD) for continuous variables and frequencies for categorical variables, were used to summarize the demographic characteristics of the study population. The Pearson correlation coefficient was then calculated to assess the relationships among the variables. Subsequently, path analysis was utilized to analyze the collected data. Path analysis is a statistical technique used to examine the relationships between a set of independent variables and a dependent variable. It extends the regression model and allows researchers to assess the fit of a causal model based on the tested correlation matrix. In this study, path analysis was employed to evaluate the relationships among the Classroom Behavior Survey, the Psychological Stress Measure (PSM), the Structural Education-School Entropy Questionnaire (QUEIs), and the Personal Teacher Entropy Questionnaire (QUEIp).

## 3. Results

Table 2 presents the means and standard deviations of the students who displayed specific behaviors in the classroom (Classroom Behavior Survey), while Table 3 shows the means and standard deviations of the PSM and Entropy questionnaires. 

Table 4 displays the Pearson’s correlations between the items of the Classroom Behavior Survey and the PSM and Entropy questionnaires. The results indicate that the perception of stress was highly correlated with the presence of children with ADHD, with a higher presence resulting in increased perceived stress (r = 0.303, *p* < 0.01). The rate of children with aggressive behavior also influenced teachers’ stress levels (r = 0.37, *p* < 0.01). On the other hand, the presence of children with math difficulties, language-related difficulties, or intellectual disabilities did not show a significant correlation with teacher stress (Table 3). 

Personal entropy was strongly and positively correlated with both the rate of students with ADHD, r (177) = 0.58, *p* < 0.01; and with children displaying aggressive behavior, r (177) = 0.36, *p* < 0.01. These findings indicate that all students with ADHD contributed to increasing teachers’ personal entropy (Table 4).

Table 5 shows the Pearson’s correlations between the subscales of the PSM and the structural and personal entropy questionnaires. In terms of the correlation between teacher stress and entropy, personal entropy showed a strong correlation with stress (r (177) = 0.68, *p* < 0.01), while structural entropy did not show a significant correlation with stress perception (r (177) = 0.221, *p* < 0.09). The subscales of the PSM that correlated with stress perception were loss of control (r (177) = 0.408, *p* < 0.01) and sense of effort and confusion (r (177) = 0.538, *p* < 0.01). 

Path analysis was conducted to estimate the relationships between variables and provide insights into the underlying causal processes. Not all hypothesized links emerged as significant from the analyses, resulting in the elimination of some, such as the presence of children with math difficulties, linguistic problems, or intellectual deficits. Figure 1 depicts the path diagram, illustrating the direct relationships between the rate of ADHD students in the classroom (β = 0.902, *p* < 0.001), the rate of children exhibiting aggression in the classroom (β = 0.404, *p* < 0.007), and personal entropy. Likewise, personal entropy has a significant positive effect on stress (β = 0.77, *p* < 0.0001). However, the path involving structural entropy was not found to be significant.

## 4. Discussion

The findings of this study validate existing literature emphasizing the significance of the teacher–student relationship within the school environment. The interaction between teachers and students is recognized as a vital factor influencing both academic and social development [1,2,3,4]. Previous studies have highlighted the specific challenges that students with ADHD present to the teacher–student relationship [12,13,14,33]. The symptoms of ADHD, characterized by inattention and hyperactivity/impulsivity, can disrupt the classroom environment and negatively affect teacher–student interactions. ADHD is a prevalent neuropsychiatric disorder among school-age children [16,17,18,19,20]. The study findings suggest that an increased frequency of students with ADHD in the classroom is associated with higher levels of entropy, indicating a less organized and more chaotic learning environment.

The study provides valuable insights into the relationship between ADHD students in the classroom, teacher entropy levels, and stress. The initial hypothesis that a high frequency of ADHD students would increase entropy levels and result in negative stress impacts on teachers was supported by the findings. The presence of ADHD symptoms was found to have a negative impact on both the structural and personal entropy levels of teachers, ultimately leading to increased stress levels.

The correlation analysis revealed that the presence of ADHD students and aggressive behavior in the classroom were significantly correlated with higher levels of perceived stress among teachers. This suggests that dealing with challenging behaviors associated with ADHD and aggression can be particularly stressful for teachers. On the other hand, the presence of students with math, language, or intellectual difficulties did not show a significant correlation with teacher stress levels. This highlights the unique challenges posed by ADHD symptoms and aggressive behaviors in the classroom.

In this study, the concept of entropy, which originates from thermodynamics and information theory, was employed to examine the connection between students with ADHD and the levels of teacher entropy. Entropy refers to the measure of disorder or uncertainty within a given system. The results suggest that an increased frequency of ADHD symptoms can lead to higher levels of disorder and entropy within the classroom system. This finding aligns with the idea that ADHD symptoms disrupt the learning environment, leading to increased chaos and disorganization [35,36,37,38]. Furthermore, the study examined two types of entropy: structural entropy and personal teacher entropy. Structural entropy refers to the physical variables that contribute to disorder in the school environment, such as classroom clutter, irregular activities, and a lack of clear rules. Personal teacher entropy encompasses subjective aspects of the teacher, including attitudes, behaviors, educational styles, and class management. Both types of entropy were found to be positively correlated with the presence of ADHD students. This suggests that ADHD symptoms can disrupt not only the physical organization of the classroom but also the subjective experiences and behaviors of teachers [20,29,50].

The findings of this study align with previous research that has highlighted the stressful nature of working with students with ADHD. Teachers have consistently rated students with ADHD as more stressful to handle, particularly when they display oppositional and aggressive behaviors. This study contributes to the existing literature by providing a deeper understanding of the relationship between ADHD students, teacher entropy levels, and stress. It underscores the need for interventions aimed at balancing the frequency of students with ADHD and promoting stress reduction strategies for teacher–student interactions.

The theoretical implications of this work are significant. By applying the concept of entropy to the classroom environment, this study offers a unique perspective on the impact of ADHD symptoms on teacher stress levels. It highlights the role of disorder and chaos in the classroom system and emphasizes the importance of maintaining a manageable level of internal entropy for effective teaching and learning. These findings contribute to our understanding of the psychological processes underlying teacher–student interactions and their implications for both teachers and students with ADHD.

The practical implications of this study are also noteworthy. The results suggest the need for interventions and support systems that address the challenges faced by teachers in classrooms with a high frequency of ADHD students. Providing training on stress reduction techniques and strategies for managing disruptive behaviors can help teachers navigate the demands of teaching students with ADHD more effectively. Creating a supportive and structured classroom environment can also contribute to reducing teacher stress and promoting positive teacher–student relationships.

One limitation of the present study is related to the assessment of classroom behavior. In this study, participant behaviors were evaluated using questionnaires, which introduced a level of subjectivity and reliance on self-reporting. Direct observation of real-time classroom behaviors was not employed, thereby limiting the ability to capture the nuanced interactions and intricacies of actual classroom dynamics. Future research could address this limitation by incorporating direct observational methods in conjunction with self-report measures. Additionally, in the Classroom Behavior Survey, teachers were asked to assess the prevalence of specific problem categories, potentially overlooking other possible issues. Therefore, it is suggested that future studies include a wider range of problem categories for a more comprehensive assessment.

Another limitation pertains to the unknown baseline stress levels of teachers prior to their interactions with children diagnosed with ADHD. To address this issue, future researchers should employ an ABA design, which involves analyzing the basal stress levels and other personality profiles of teachers before the commencement of the school term (A1) and after two months of interaction with the children (A2). This would provide valuable insights into the potential impact of teacher–student interactions on teacher stress levels and overall wellbeing.

All these suggestions would provide a more comprehensive and objective understanding of the phenomenon of classroom behavior. 

Further investigation can delve deeper into the factors that contribute to the structural and personal entropy of teachers. This may involve assessing the physical conditions of the classroom, teachers’ classroom management practices, and their emotional wellbeing. Also useful may be the exploration of the classroom environment: exploring the impact of classroom environmental factors such as classroom layout, organization, and visual stimuli on teacher entropy levels and stress. This could provide insights into how modifying the physical environment can contribute to reducing entropy and improving teacher wellbeing. Moreover, future research can analyze the investigation of teacher coping strategies: investigating the coping strategies employed by teachers when dealing with the challenges posed by students with ADHD. This could include analyzing the effectiveness of different coping mechanisms, such as mindfulness practices, relaxation techniques, or professional development programs focused on stress management.

Future longitudinal studies can examine the long-term impact of teacher stress and entropy levels on their job satisfaction, retention rates, and overall professional development. This would provide a comprehensive understanding of the sustained effects of working with students with ADHD on teachers’ lives. Moreover, the comparison of teacher entropy across educational settings, such as mainstream classrooms, specialized ADHD support programs, or inclusive education settings, could shed light on how varying contexts influence teacher entropy and inform the development of targeted interventions.

## 5. Conclusions

In conclusion, this study demonstrates that the presence of ADHD symptoms in the classroom has a negative impact on teacher entropy levels and significantly increases stress levels. The findings highlight the disruptive nature of ADHD symptoms and their implications for teacher wellbeing. The study emphasizes the need for interventions that promote a balanced classroom environment and provide teachers with the necessary tools to manage the challenges associated with ADHD. By addressing the stressors and entropy levels in the classroom, interventions can contribute to improving teacher–student interactions and ultimately enhance the educational experience for both teachers and students.

## Figures and Tables

**Figure 1 children-10-01082-f001:**
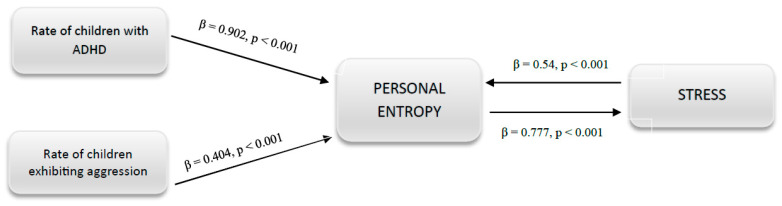
Path diagram.

**Table 1 children-10-01082-t001:** Means (M) and standard deviations (SD) of the samples.

Measures	M	SD
Teachers		
Number of males, females, not specified	17, 156, 4	
Age	43.76	7.95
Years of teaching	19.78	11.03
Children		
Number of males, females, not specified	869, 1169, 301	
Age	7.92	1.8
Average class number	24.1	3.01

**Table 2 children-10-01082-t002:** Means (M) and standard deviations (SD) of the Classroom Behavior Survey for each class (with a sample mean of 24.1 ± 3.01).

Measures	M	SD
ADHD	1.25	1.02
Aggressive behavior	0.30	0.95
Problems in the linguistic area	0.81	0.93
Problems in the mathematical area	0.92	1.01
Intellectual disability	0.052	0.73

**Table 3 children-10-01082-t003:** Means (M) and standard deviations (SD) for each of the subscales of MSP and entropy scales.

Measures	M	SD
MSP		
Loss of control, irritability	1.77	0.53
Psychophysiological sensations	1.35	0.47
Sense of effort and confusion	1.57	0.56
Depressive anxiety	1.55	0.58
Pains and physical problems	1.44	0.59
Hyperactivity	2.17	0.61
Entropy		
Structural	45.54	6.11
Personal	66.74	8.08

**Table 4 children-10-01082-t004:** Pearson’s correlations between structural and personal entropy, and stress.

Measures	Structural Entropy	Personal Entropy	Stress
ADHD	0.095	0.58 **	0.303 **
Aggressive behaviors	0.352 *	0.368 **	0.374 **
Problems in the linguistic area	0.089	0.015	0.265
Problems in the mathematical area	0.021	0.011	0.192
Intellectual disability	0.319	0.201	0.155

* *p* < 0.05; ** *p* < 0.01.

**Table 5 children-10-01082-t005:** Pearson’s correlations between structural entropy, personal entropy, and stress subscales.

Stress Subscales	Structural Entropy	Personal Entropy
Loss of control	−0.121	0.408 **
Psychophysiological sensations	0.071	0.321 *
Sense of effort and confusion	0.146	0.538 **
Depressive anxiety	0.001	0.288
Pains and physical problems	−0.055	0.152
Hyperactivity	−0.178	0.315 *
Total stress	0.221	0.68 **

* *p* < 0.05; ** *p* < 0.01.

## Data Availability

Not applicable.

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
