# Peer review of "ADHD Symptoms Increase Perception of Classroom Entropy and Impact Teacher Stress Levels"

_children, 2023, doi:10.3390/children10061082_

Round 1

Reviewer 1 Report

The Authors of this manuscript explore the impact of negative stress on teachers who are interacting with students diagnosed with ADHD, from the perspective of internal and external entropy, which is an interesting and original approach. There are some points that need to be addressed, please see below:

Abstract

-Consider replacing “impacts” with “impact”;

Methods

-lines 104-105: “The children in this study were 2339…” is not a valid formulation, please consider rephrasing, maybe “This study enrolled 2339 children distributed…”; were all 2339 children diagnosed with ADHD? If they were mixed with clinically normal children in the same classrooms, please specify the percentages for each category; Were any children diagnosed with learning disabilities or another psychiatric diagnostic in this group?; Were any other relevant variables controlled in this study, e.g., children undergoing pharmacological treatment or psychological counseling? What about the basal stress level in teachers before their interaction with children diagnosed with ADHD, or their personality profile?; if they were not controlled for, consider mentioning these in the “Limitations” section; line 108- consider replacing “much” with “many”;

-were the teachers informed which students were diagnosed with ADHD in a certain classroom? Or did they know only that those students were under psychiatric observation?

-were all the instruments used standardized on the Italian population? It is important to mention whether QUEIs and QUEIp were created by Italian researchers or only adapted by them;

-lines 189-191- there is a misalignment of these lines, please correct this;

-were the children aware of the fact that their behavior was evaluated by their teachers during this study?

-any suggested future directions of research in this field?

Tables

-Table 1: please consider re-formatting this table, it is quite hard to comprehend its content, e.g., M and SD are on the lines, but also on the columns; please define “M”, “nd”, and “SD”, as you did in the other tables; consider splitting the lines including values for male, female and non-disclosed, it is difficult to understand those values if they are just separated by commas; “mail” is a typo;

Minor editing of English language is needed.

Author Response

First referee

The Authors of this manuscript explore the impact of negative stress on teachers who are interacting with students diagnosed with ADHD, from the perspective of internal and external entropy, which is an interesting and original approach. There are some points that need to be addressed, please see below:

Point 1:

Abstract

-Consider replacing “impacts” with “impact”;

Response 1: thank you, we replaced it.

Point 2:

Methods

-lines 104-105: “The children in this study were 2339…” is not a valid formulation, please consider rephrasing, maybe “This study enrolled 2339 children distributed…”; were all 2339 children diagnosed with ADHD? If they were mixed with clinically normal children in the same classrooms, please specify the percentages for each category; Were any children diagnosed with learning disabilities or another psychiatric diagnostic in this group?; Were any other relevant variables controlled in this study, e.g., children undergoing pharmacological treatment or psychological counseling? What about the basal stress level in teachers before their interaction with children diagnosed with ADHD, or their personality profile?; if they were not controlled for, consider mentioning these in the “Limitations” section; line 108- consider replacing “much” with “many”;

Response 2:

Thank you. We have addressed the first part regarding the sample as follows: This study included a total of 2,339 children, who were enrolled from 177 classes in primary public schools located in Milan and Messina and their respective provinces. The age range of the children varied from 6 to 10 years old. The sample encompassed all the children within each classroom, comprising both clinically normal children and children with clinical conditions. All clinically significant children had received an official diagnosis from healthcare facilities in their respective regions. Among the sample, 4.98% exhibited symptoms of Attention-Deficit/Hyperactivity Disorder (ADHD), 1.2% displayed aggressive behaviors, 3.2% had language disorders, 3.6% experienced difficulties in mathematics, and 2% presented intellectual disabilities.

We added also the limitations in the final part as follows:

Additionally, in the classroom behavior survey, teachers were asked to assess the prevalence of specific problem categories, potentially overlooking other possible issues. Therefore, it is suggested that future studies include a wider range of problem categories for a more comprehensive assessment. Another limitation pertains to the unknown baseline stress levels of teachers prior to their interactions with children diagnosed with ADHD. To address this issue, future researchers should employ an ABA design, which involves analyzing the basal stress levels and other personality profiles of teachers before the commencement of the school term (A1) and after two months of interaction with the children (A2). This would provide valuable insights into the potential impact of teacher-student interactions on teacher stress levels and overall well-being.

Point 3:

-were the teachers informed which students were diagnosed with ADHD in a certain classroom? Or did they know only that those students were under psychiatric observation?

Response 3:

The teachers were informed about the specific students who were diagnosed with ADHD in their respective classrooms. In Italy, it is common practice for healthcare institutions to not only communicate the diagnosis to the parents but also to share this information, with parental consent,  with the teachers. This allows for the necessary support to be provided to the clinically significant groups within the classroom. In order to capture this information, we directly asked the teachers to report the number of students experiencing difficulties in the classroom behavior survey. In this study, teachers were aware of the diagnoses of the students, enabling them to have a better understanding of the specific challenges and needs within their classroom setting. This information was valuable for assessing and addressing the behavioral characteristics associated with ADHD and other clinical conditions. In the paragraph 2.2.1. Classroom Behavior Survey it was specified the following:

“It was emphasized that the teachers should only report students who had received official diagnoses from specialized physicians, excluding any subjective opinions or beliefs regarding the difficulties of the children. The focus was on reporting the actual number of students who displayed these specific behaviors in their classroom”.

Point 4:

-were all the instruments used standardized on the Italian population? It is important to mention whether QUEIs and QUEIp were created by Italian researchers or only adapted by them;

Response 4:

All the instruments were standardized in the Italian population. Moreover we added the requested information in the text. The first work of Fabio et al.(2005) created the instruments and submitted it in the Italian population, the second work (Pinnelli and De Nitto, 2018) confirmed the psychometric properties again in an Italian sample. We added this information in the text.

Point 5:

-lines 189-191- there is a misalignment of these lines, please correct this;

Response 5:

Thank you. We corrected it.

Point 6:

--were the children aware of the fact that their behavior was evaluated by their teachers during this study?

Response 6:

The children were not aware of the evaluation.

Point 7:

-any suggested future directions of research in this field?

Response 7:

We added them as follows:

Further investigation can delve deeper into the factors that contribute to the struc-tural and personal entropy of teachers. This may involve assessing the physical condi-tions of the classroom, teachers' classroom management practices, and their emotional well-being. It may be also useful the exploration of classroom environment: Explore the impact of classroom environmental factors, such as classroom layout, organization, and visual stimuli, on teacher entropy levels and stress. This could provide insights into how modifying the physical environment can contribute to reducing entropy and im-proving teacher well-being. Moreover, the future research can analyze the investiga-tion of teacher coping strategies: Investigate the coping strategies employed by teachers when dealing with the challenges posed by students with ADHD. This could include analyzing the effectiveness of different coping mechanisms, such as mindfulness prac-tices, relaxation techniques, or professional development programs focused on stress management. Future longitudinal studies can examine the long-term impact of teacher stress and entropy levels on their job satisfaction, retention rates, and overall professional devel-opment. This would provide a comprehensive understanding of the sustained effects of working with students with ADHD on teachers' lives. Moreover, the comparison of teacher entropy across educational settings, such as mainstream classrooms, specialized ADHD support programs, or inclusive education settings. This could shed light on how varying contexts influence teacher entropy and inform the development of targeted in-terventions.

Point 8:

Tables

-Table 1: please consider re-formatting this table, it is quite hard to comprehend its content, e.g., M and SD are on the lines, but also on the columns; please define “M”, “nd”, and “SD”, as you did in the other tables; consider splitting the lines including values for male, female and non-disclosed, it is difficult to understand those values if they are just separated by commas; “mail” is a typo;

Response 8:

We re-formatted it.

Comments on the Quality of English Language

Minor editing of English language is needed.

Thank you. We did it.

Reviewer 2 Report

Hello Dears;

Thank you for your practical research project.

Comments:

-A more comprehensive explanation should be given about the validation and localization of QUEIs, QUEI p, CBS questionnaires in the Italian society.

-It should be clarified that the diagnosis of ADHD was made only with a questionnaire or it was also confirmed with a clinical interview based on the diagnostic criteria of DSM-5.

-Was the study one-sided blind or not? were the researcher and the data analyst different or not?

Author Response

Second Referee

Hello Dears;

Thank you for your practical research project.

Comments:

Point 1:

-A more comprehensive explanation should be given about the validation and localization of QUEIs, QUEI p, CBS questionnaires in the Italian society.

Response 1:

All the instruments were standardized in the Italian population. The first work of Fabio et al. (2005) created the instruments and submitted them to the Italian populations in the northern and southern regions of Italy. The second work (Pinnelli and De Nitto, 2018) confirmed the psychometric properties once again, this time using an Italian sample from the southern regions of Italy. We have added this additional information into the text.

Point 2:

-It should be clarified that the diagnosis of ADHD was made only with a questionnaire or it was also confirmed with a clinical interview based on the diagnostic criteria of DSM-5.

Response 2:

In Italy, it is common practice for healthcare institutions to communicate the diagnosis to the parents and, with parental consent, share this information with the teachers. This approach ensures that necessary support can be provided to clinically significant groups within the classroom. To capture this information, we directly requested teachers to report the number of students experiencing difficulties in the classroom behavior survey. In this study, teachers were aware of the diagnoses of the students, enabling a comprehensive assessment of the situation. This information was valuable for assessing and addressing the behavioral characteristics associated with ADHD and other clinical conditions. In the paragraph 2.2.1. Classroom Behavior Survey it was specified the following:

“It was emphasized that the teachers should only report students who had received official diagnoses from specialized physicians, excluding any subjective opinions or beliefs regarding the difficulties of the children. The focus was on reporting the actual number of students who displayed these specific behaviors in their classroom”.

Point 3:

-Was the study one-sided blind or not? were the researcher and the data analyst different or not?

Response 3:

Thank you. No, the study was not one-sided blind as the teachers were aware of the diagnoses of the students. Additionally, the researcher and the data analyst were different individuals.

Reviewer 3 Report

Hello.

Interesting study, but I have questions.

1. You cite your work very often, I counted more than 5 references. How do you explain the need for such a mention to your research. It looks like an artificial citation boost.

2. You need to add a null hypothesis.

3. It is necessary to reconsider the objective attitude towards social networks, the validity of which is not registered anywhere

Author Response

Third Referee

Point 1:

  1. You cite your work very often, I counted more than 5 references. How do you explain the need for such a mention to your research. It looks like an artificial citation boost.

Response 1:

We deleted two works: 16. Fabio, R. A., Towey, G. E., & Caprì, T. Static and dynamic assessment of intelligence in ADHD subtypes. Frontiers in Psychology 2022, 13 doi:10.3389/fpsyg.2022.846052.

  1. Fabio, R.A., Andricciola, F. & Caprì, T. Visual-motor attention in children with ADHD: The role of automatic and controlled processes. Research in Developmental Disabilities 2022, 97 doi:10.1016/j.ridd.2022.104193.

Point 2:

  1. You need to add a null hypothesis.

Response 2:

Thank you. We added the null hypothesis

Point 3:

  1. It is necessary to reconsider the objective attitude towards social networks, the validity of which is not registered anywhere

Response 3:

I think it is useful to reconsider the objective attitude towards social networks, but I'm not clear on how it can be useful in this context.

Round 2

Reviewer 1 Report

The manuscript improved significantly